# Magnetite Nanoparticle Assemblies and Their Biological Applications: A Review

**DOI:** 10.3390/molecules29174160

**Published:** 2024-09-02

**Authors:** Jinjian Wei, Hong Xu, Yating Sun, Yingchun Liu, Ran Yan, Yuqin Chen, Zhide Zhang

**Affiliations:** 1College of Chemistry, Chemical Engineering and Materials Science, Shandong Normal University, Jinan 250014, China; 2023300228@stu.sdnu.edu.cn (H.X.); 202210300123@stu.sdnu.edu.cn (Y.S.); zdzhang@sdnu.edu.cn (Z.Z.); 2Jinan Guoke Medical Technology Development Co., Ltd., Jinan 250000, China; liuyingchun@sibet.ac.cn; 3Jinan Petrochemical Design Institute, Jinan 250100, China; yanran1108@126.com

**Keywords:** magnetite nanoparticle, self-assembly, bottom up, biological applications

## Abstract

Magnetite nanoparticles (Fe_3_O_4_ NPs) have garnered significant attention over the past twenty years, primarily due to their superparamagnetic properties. These properties allow the NPs to respond to external magnetic fields, making them particularly useful in various technological applications. One of the most fascinating aspects of Fe_3_O_4_ NPs is their ability to self-assemble into complex structures. Research over this period has focused heavily on how these nanoparticles can be organized into a variety of superstructures, classified by their dimensionality—namely one-dimensional (1D), two-dimensional (2D), and three-dimensional (3D) configurations. Despite a wealth of studies, the literature lacks a systematic review that synthesizes these findings. This review aims to fill that gap by providing a thorough overview of the recent progress made in the fabrication and organization of Fe_3_O_4_ NP assemblies via a bottom-up self-assembly approach. This methodology enables the controlled construction of assemblies at the nanoscale, which can lead to distinctive functionalities compared to their individual counterparts. Furthermore, the review explores the diverse applications stemming from these nanoparticle assemblies, particularly emphasizing their contributions to important areas such as imaging, drug delivery, and the diagnosis and treatment of cancer.

## 1. Introduction

Magnetic nanoparticles (MNPs) are usually categorized as ferromagnetic and ferrimagnetic materials. The composition of ferromagnetic nanoparticles can be elemental [1,2,3,4], alloy [5], oxide [6], or composite forms of substances primarily comprising iron (Fe), cobalt (Co), and nickel (Ni) [7]. In contrast to ferromagnetic substances, ferrimagnetic materials are typically dual oxides composed of iron and another metal [8]. The ferrimagnetic materials most commonly used in technical devices are known as ferrites. These ferrites are cubic or hexagonal shape. Cubic ferrites have a chemical formula of MO·Fe_2_O_3_, where M represents a divalent ion such as Mn^2+^, Fe^2+^, Co^2+^, or Ni^2+^. In recent years, these materials have garnered significant attention due to their various applications, particularly in fields such as imaging [9], cancer therapy [10,11,12], and drug delivery [13,14], among others [15,16]. Among the different types of MNPs, magnetite nanoparticles (Fe_3_O_4_ NPs) stand out due to their excellent biocompatibility, which allows them to integrate seamlessly into biological systems. They are naturally found in living organisms such as magnetic bacteria [17,18], migratory ants [19,20], fish [21], and birds [22]. A very interesting example of the mid-range of living organisms are migratory ants, in which a nanomagnetite-based one dimensional arrangement is used as a compass for navigation while escaping from predators and searching for food. These enhance their appeal for biomedical applications. Over the last couple of decades, numerous synthetic approaches have been developed to produce high-quality Fe_3_O_4_ NPs. These methods include chemical synthesis techniques such as thermal decomposition [23,24], hydrothermal reaction [25], coprecipitation [26], and sol-gel processes [27], all of which contribute to the effective generation of these nanoparticles. The magnetic characteristics of Fe_3_O_4_ NPs are significantly affected by their size and shape [15]. Experimental observations have indicated that larger Fe_3_O_4_ NPs tend to have a higher lateral relaxation rate (r_2_) relaxivity in the size range of 4 to 12 nm [28]. This phenomenon is believed to arise from the increase in the magnetization of nanoparticles as their size expands. Interestingly, for nanoparticles exceeding the size of 12 nm, the r_2_ relaxivity does not continue to rise with increasing the size. This particular size region is referred to as the static dephasing regime (SDR) [29,30]. In the SDR, the magnetic field produced by nanoparticles is sufficiently strong that the diffusion of water has minimal impact on the T_2_ relaxation process. Consequently, it is anticipated that a plateau in the maximum r_2_ will emerge in SDR. Beyond this specific threshold, the relaxivity begins to decrease as the size of nanoparticles increases further. This counterintuitive behavior highlights the complex relationship between the size of Fe_3_O_4_ NPs and their magnetic properties, which is crucial for tailoring their applications in various biomedical and industrial fields.

In comparison to the isolated Fe_3_O_4_ NPs, their assemblies possess unique relaxivity characteristics, as their specific nanostructures change the proton relaxation phenomenon [31,32]. Taking clusters as an example, a collection of magnetic nanoparticles can theoretically be visualized as a large sphere with a magnetic charge, where its total magnetic moment correlates with the size [15]. Consequently, Fe_3_O_4_ NP clusters demonstrate a greater magnetic moment in comparison to dispersed Fe_3_O_4_ NPs, and the transverse relaxivity of the assemblies is influenced by their magnetization levels [15]. In light of these observations, the assemblies offer enhanced sensitivity of detection in magnetic resonance imaging. The ability to self-assemble these nanoparticles is a critical area of research, particularly because surface modification using ligands with controlled interaction properties plays a key role in this process. The overall structure of these NP assemblies is largely dictated by the delicate balance between attractive and repulsive forces acting between individual particles [33]. This balance can be meticulously adjusted through the use of surface-coated ligands, thereby allowing for the fine-tuning of assembly characteristics. Studies have concentrated on specific types of ligands, particularly those that are terminated with amino [34], carboxylic acid [35,36,37], and catechol groups [38,39]. These ligands are favored for their distinct hydropathy properties and their strong affinities for the surfaces of Fe_3_O_4_ NPs. Effectively controlling colloidal interactions, including magnetic dipole-dipole forces, van der Waals interactions, electrical double-layer steric effects, and both hydrophobic and solvation forces, is essential for achieving particle stabilization [40,41]. Nevertheless, it should be emphasized that most of the documented Fe_3_O_4_ NPs have been produced in organic solvents [23,24,25,26,27]. This reliance on the organic media limits their dispersibility in aqueous solutions, which poses a challenge for direct biological applications due to the inherent hydrophobicity of original ligands. To overcome these limitations, employing bottom-up approaches in the self-assembly of Fe_3_O_4_ NPs presents an efficient strategy for creating assemblies that are readily dispersible in water [42,43]. The recent advancements in the self-assembly techniques for Fe_3_O_4_ NPs hold considerable significance, particularly in broadening the array of prospective applications within biological contexts. These developments pave the way for innovative uses of these nanoparticles, enhancing their utility in various biological applications. This review seeks to underscore the notable progress made in this area, focusing on the self-assembly methods utilized for Fe_3_O_4_ NPs and exploring their potential impact across diverse biological settings. By examining these advancements, we aim to elucidate how they can be leveraged effectively in practical applications, thereby contributing to the advancement of the biological field (Figure 1).

## 2. Self-Assembly of Fe_3_O_4_ NPs

Self-assembly of Fe_3_O_4_ NPs is an intriguing process where these particles organize themselves into structured arrangements through non-covalent interactions [31,32,33,34,35,46,47]. By modifying the surface of NPs with polymers or functional molecules, the interactions between particles can be changed, thus influencing the self-assembly [32,37,42]. In addition to these, the inherent magnetic dipole–dipole interactions are also important for the self-assembly of Fe_3_O_4_ NPs, especially when a magnetic field is applied [48,49]. Compared with individual Fe_3_O_4_ NPs, their assemblies show a distinctive magnetic response [42,50]. Studies have shown that the assemblies can effectively enhance T_2_ relaxation and result in a higher r_2_ value, improving magnetic response while maintaining their superparamagnetic properties [32,42,43]. Up to now, research has classified the self-assembly of Fe_3_O_4_ NPs into various structures including one-dimensional (1D), two-dimensional (2D), and three-dimensional (3D) structures.

### 2.1. 1D Fe_3_O_4_ NP Nanoarrays

In the natural world, Fe_3_O_4_ NPs possess a remarkable capability for spontaneous self-assembly into 1D chains, particularly observed in magnetotactic bacteria [17,18]. These unique microorganisms have evolved the ability to exploit Earth’s geomagnetic field, which they use as a guide for orientation and navigation through their environment. This fascinating interplay between biology and magnetism illustrates the innovative adaptation of these bacteria to their surroundings. Beyond the natural occurrences of self-assembly in magnetotactic bacteria, it is possible to replicate the formation of 1D Fe_3_O_4_ NP chains through engineered bottom-up self-assembly techniques [49,50,51,52,53,54]. This process can be facilitated either under the existence of external magnetic field or through other methods that do not require such a field. The ability to manage and alter the magnetic characteristics of nanoparticle chains is crucial, as it can be affected by different elements such as the dimensions and makeup of the nanoparticles, along with the length of resulting structures. Studies have shown that the formation of elongated chains from Fe_3_O_4_ NPs can provide numerous advantages in the medical and imaging fields [49].

The controlled self-assembly of Fe_3_O_4_ NPs into 1D chains can be effectively achieved through the strategic combination of azide and alkyne “click” reactions, complemented by the application of an external magnetic field (Figure 2) [49]. In this process, the surface of nanoparticles is functionalized with azide-terminated ligands, referred to as NP@N_3_. Simultaneously, alkyne-terminated self-assembled monolayers (SAM-CC) are anchored to gold substrates. The “click” reaction occurs in a solution containing both NP@N_3_ and SAM-CC, facilitated by the catalysts such as triethylamine and CuBr(PPh_3_)_3_. Underneath the substrates, the magnet system generates an in-plane magnetic field, which induces uniaxial shape anisotropy in the assemblies. Additionally, this magnetic field plays a crucial role in slowing down the kinetics of self-assembly process, ultimately leading to the formation of chain-like structures. These chain structures demonstrate enhanced collective magnetic properties in comparison to their dispersed Fe_3_O_4_ NP counterparts, a phenomenon attributable to the intrachain dipolar interactions present within the assembled chains. An alternative approach for fabricating 1D chain-like structures involves the self-assembly of amphipathic polymers and Fe_3_O_4_ NPs at the air–water interface [50]. In scenarios devoid of an external magnetic field, the self-assembly of poly(3-hexylthiopene)-block-poly(ethylene glycol) (P3HT-b-PEG) alongside Fe_3_O_4_ NPs results in the creation of 1D nanowires. This occurrence is primarily due to the inherent capability of P3HT-b-PEG to form polymeric nanowires on its own. During this assembly, Fe_3_O_4_ NPs are arranged along the periphery of P3HT nanowires at the interface of P3HT and PEG, facilitating the formation of organized 1D arrays. The resulting P3HT-b-PEG/ Fe_3_O_4_ NP arrays exhibit random orientations at various positions within the film. Upon the application of an external magnetic field, these nanowires remain able to form, yet their alignment tends to become nearly parallel to the direction of magnetic field over a sub-millimeter scale. This study demonstrates that magnetic field-induced self-assembly methods can be successfully employed to create polymer films exhibiting both macroscopic order and nanometer-scale structure, even under low magnetic field conditions. In another report, 1D Fe_3_O_4_ NP arrays were controllably generated using tartrate- and polyaspartate-coated colloidal Fe_3_O_4_ NPs through the application of an external magnetic field [55]. It was found that larger Fe_3_O_4_ NPs enhance the magnetic dipole interaction when compared with their smaller counterparts. Polyaspartate, which is longer than tartrate surfactant molecules, assures the stability with smaller aggregates. Theoretical analysis has suggested that the average chain length tends to rise as both the aging time and particle volume fraction increase. As the strength of applied magnetic field increases, single chains of Fe_3_O_4_ NPs can merge together building columnar-like nanostructures in a phase transition field-induced process, thus adding extra complexity and richness to the 1D nanoarrays. One-dimensional nanoarrays can also be generated independently of an external magnetic field [48]. Specifically, Fe_3_O_4_ NPs possess the ability to self-assemble into 1D chains, both on substrates and within colloidal dispersions, through interactions driven by magnetic dipoles, thereby eliminating the necessity for external magnetic influences. To enhance these structures, 1D assemblies with a silica shell surface were subsequently fabricated by employing a sol–gel process. With the introduction of a magnetic field, the resulting nanostructures exhibited the capacity to orient and align themselves according to the direction of that external magnetic field, signifying a controlled method of manipulation for these nanoarrays.

### 2.2. 2D Fe_3_O_4_ NP Nanoarrays

The properties exhibited by nanoparticle assemblies are significantly influenced by the specific motif of the assembly. In particular, dipolar interactions have been found to exert a more pronounced effect on 2D arrays than on powdered samples [56,57,58,59,60,61,62,63,64,65]. This difference arises from the shape anisotropy introduced when nanoparticles are organized into thin films, which enhances the strength and significance of these interactions. Moreover, dipolar interactions can influence the assembly process, leading to the creation of densely packed nanostructures. This characteristic poses a challenge to regulating the distances between individual nanoparticles on surfaces, complicating efforts to manipulate the arrangement and spatial distribution of these nanoparticles.

A monolayer of Fe_3_O_4_ NPs can be synthesized at the interface between heptane and diethylene glycol through a self-limited self-assembly mechanism [56]. Notably, after the formation of the initial monolayer, nanoparticles did not progress to form a multilayer structure, despite the presence of excess of nanoparticles dispersed in the upper heptane phase. The presence and stability of this monolayer were confirmed using real-time optical reflection measurements of incident polarized light, particularly at Brewster’s angle, which is a critical optical phenomenon that indicates the unique properties of the film at the interface. Similarly, Fe_3_O_4_ NPs modified with polyethylene glycol (PEG) could facilitate their self-assembly into 2D films at the interface between air and aqueous suspensions [58]. This process was influenced by the control of salt concentration in the solution. The introduction of potassium carbonate prompted the PEG-Fe_3_O_4_ NPs to migrate towards the liquid/vapor interface, resulting in the formation of a single layer. As the salt concentration and/or the concentration of nanoparticles increased, the organization of nanoparticles adsorbed on the surface improved. However, additional increase in salt concentration resulted in the emergence of a second, albeit incomplete, layer of nanoparticles adjacent to the top-most layer, which subsequently initiated a precipitation process.

Experimental results have shown that the self-assembly process of Fe_3_O_4_ NPs that have been functionalized with fatty acids or dendritic molecules is capable of producing uniform and high-density 2D arrays. This impressive organizational ability of nanoparticles was achieved through the application of the Langmuir–Blodgett technique, which facilitates the orderly arrangement of these functionalized nanoparticles on a planar surface [59]. The interparticle distance within these arrays could be systematically adjusted by managing the size of the molecules that coated the surfaces of nanoparticles. Furthermore, the dipolar interactions observed in the 2D arrays were found to be stronger than those in 3D powdered samples, indicating that the dimensionality of the assembly plays a crucial role in the strength of these interactions. Additionally, the formation of Janus-structured nanoparticles was identified as a critical factor driving the self-assembly process utilizing Langmuir–Blodgett technique. The large-scale fabrication of monolayer films composed of Fe_3_O_4_ NPs using a vacuum deposition technique can also be fabricated (Figure 3) [61]. During the deposition process, the self-assembly behavior of nanoparticles was primarily influenced by the interactions occurring between nanoparticles themselves. Both cubic and spherical Fe_3_O_4_ NP monolayer films exhibited strong magnetic anisotropy, which is attributed to the inter-particle dipolar interactions between nanoparticles, effectively enhancing their magnetic properties. In another report, a different approach for the self-assembly of Fe_3_O_4_ NPs, combining electrophoresis with template-assisted techniques, was explored [62]. Applying an electric field, in conjunction with patterned substrates, facilitates the selective nucleation of self-assembled nanoparticle arrays along the edges of hydrogen silsesquioxane patterns. The analysis of interaction energies indicated that the contributions of electrostatic and capillary forces to nanoparticle self-assembly corroborate the hypothesis that the electric field serves not only to increase particle concentration but also to guide the particles to specific nucleation sites. This selective nucleation capability enables the formation of intricate nanoparticle chains and arrays that can be tailored to the geometry of underlying patterns, allowing for the creation of chain formations that may include shapes such as rings or squares. Morais and coworkers reported the reconstruction of pre-fabricated 2D films comprising iron oxide (e.g., maghemite) NPs deposited onto flat glass surfaces [63]. The reconstruction of 2D arrays of maghemite NPs immersed in dimercaptosuccinic acid (DMSA) solutions at increasing molar concentration was investigated. Atomic force microscopy micrographs provide strong evidence of different mechanisms for 2D reconstruction, depending on the DMSA concentration and aging time. In a recent study, the self-assembly of Fe_3_O_4_ nanocubes featuring a hydrophobic surface resulted in the formation of flux-closure polygons via a drop-casting method [65]. The creation of nanogons occurred exclusively under conditions where there was limited exposure to magnetic fields throughout the synthesis of the nanocubes. Furthermore, it was noted that the substrate used in the process had to be hydrophilic. This study concluded that both magnetostatic forces and hydrophobic interactions play a crucial role in facilitating the formation of these nanogons.

### 2.3. 3D Fe_3_O_4_ NP Superstructure

The development of a 3D superstructure composed of Fe_3_O_4_ NPs significantly enhances the effectiveness of isolated NPs through a phenomenon known as the collective effect [66,67]. When Fe_3_O_4_ NPs undergo self-assembly to form clusters, their relaxation rate experiences an increase, which subsequently leads to improved performance in magnetic resonance imaging (MRI). This enhancement parallels that observed with individual Fe_3_O_4_ NPs, as the clusters also exhibit three distinct regimes of r_2_ relaxation rates that vary in size [68]. Initially, there is an increase in r_2_ value within the motional average regime (MAR), which is followed by a peak or plateau in the SDR, and finally a decrease in r_2_ is observed in the echo-limiting regime (ELR). It is important to note that SDR for Fe_3_O_4_ NP clusters reaches around 100 nm, which underscores the necessity for carefully controlling the aggregation of these nanoparticles [69,70]. Achieving optimal aggregation is essential for maximizing the r_2_ relaxivity, thereby ensuring that the clusters perform at their best in diagnostic imaging applications. Hence, understanding and manipulating the self-assembly process of Fe_3_O_4_ NPs is key for enhancing their effectiveness in medical imaging, ultimately leading to improved diagnostic capabilities.

To date, a range of protocols has been established for creating clusters of magnetic nanoparticles. Considerable focus has been placed on directly synthesizing clusters that are water-dispersible and specifically consist of Fe_3_O_4_ NPs [67,71,72,73]. Xie et al. documented the emergence of Fe_3_O_4_ NP clusters of varying sizes and configurations. This was accomplished through the self-assembly of amphiphilic mPEG-PLA copolymer in conjugation with hydrophobic Fe_3_O_4_ NPs in an aqueous environment [74]. The study revealed that the proportion of copolymer to Fe_3_O_4_ NPs was vital for the successful formation of these clusters. Furthermore, it was observed that the transverse relaxivity of condense Fe_3_O_4_ NP clusters exhibited a dependence on cluster size. By increasing the size of clusters, the T_2_ relaxation rate gradually increases. In addition to the aforementioned method, the oil-in-water emulsion method can also be used to fabricate NP clusters [75]. With this methodology, hydrophobic Fe_3_O_4_ NPs are initially dispersed within an organic phase. Subsequently, an aqueous phase containing hexadecyltrimethyl ammonium bromide is introduced, resulting in the formation of oil-in-water emulsion. Upon complete evaporation of the organic phase, nanoparticles aggregate within the micelles, culminating in the formation of cluster structures. This particular strategy not only achieves a high yield of nanoparticle cluster production but also facilitates the manipulation of surface chemical properties of the resulting nanoclusters.

Fe_3_O_4_ nanoparticle vesicles (NVs) with a hollow interior also exhibit excellent magnetic properties and response characteristics [76,77,78,79,80]. Compared to nanoparticle clusters, their hollow interior allows for the encapsulation of a greater quantity of target drugs. To date, the primary strategy for fabricating Fe_3_O_4_ NVs involves the incorporation of hydrophobic Fe_3_O_4_ NPs into the membranes of liposomes or polymeric vesicles [76,77,78,79]. However, the contrast in MRI is influenced by the size and content of Fe_3_O_4_ NPs within the assembly, as well as the morphology of assemblies. In the case of liposome vesicles, to prevent changes in morphology, the size of incorporated Fe_3_O_4_ NPs is typically kept below 8 nm, resulting in a relatively small number of embedded Fe_3_O_4_ NPs. Consequently, the relative content of Fe_3_O_4_ NPs remains low, leading to unsatisfactory MRI outcomes. As a result, the current research predominantly focuses on utilizing block copolymers to modify Fe_3_O_4_ NPs, facilitating their cooperative self-assembly into Fe_3_O_4_ NVs [77,78,79,80]. Park et al. discovered that by adjusting the solvent and amphiphilic block copolymers, the interaction between copolymer and Fe_3_O_4_ NPs resulted in the creation of magnetic micelles, core-shell structures, and NVs with densely arranged surface Fe_3_O_4_ NPs [77]. Among these structures, Fe_3_O_4_ NVs exhibited the highest r_2_ value due to the increased density of embedded Fe_3_O_4_ NPs and the enhanced permeability of water molecules. Furthermore, they found that the r_2_ value of Fe_3_O_4_ NVs increased by increasing the size of Fe_3_O_4_ NPs in the membrane, but decreased by increasing the size of Fe_3_O_4_ NVs (Figure 4a–d) [78]. Nie and coworkers demonstrated that Fe_3_O_4_ NPs modified with catechol-terminated block copolymers, along with free styrene-acrylic acid block copolymer (PS-b-PAA), can self-assemble into Fe_3_O_4_ NVs with closely arranged surface Fe_3_O_4_ NPs (Figure 4e,f) [80]. The thickness of vesicle membrane can be controlled by adjusting the weight ratio of PS-b-PAA to Fe_3_O_4_ NPs, leading to the formation of single-, double-, and multi-layer membrane NVs composed of Fe_3_O_4_ NPs. Notably, multi-layer NVs exhibit the highest r_2_ value, indicating their potential for use in magnetic resonance imaging-guided cancer treatment.

## 3. Application of Fe_3_O_4_ NP Assemblies

The assembly of Fe_3_O_4_ NPs has enhanced magnetic properties compared to the monodispersed Fe_3_O_4_ NPs [27,32,42,81,82]. Therefore, its assembly has applications based on the monodispersed Fe_3_O_4_ NPs. It also shows enhanced effects and has a wide range of potential applications in bioimaging [44,83], cancer diagnosis, and treatment [45,84,85,86], drug delivery [80,86].

### 3.1. Imaging

MRI and ultrasound imaging (USI) are two complementary medical imaging techniques, with MRI offering high spatial resolution and USI providing high sensitivity. To enhance the signal-to-noise ratio of images, both MRI and USI require the use of contrast agents. Assemblies of Fe_3_O_4_ NPs with superparamagnetic properties can be utilized to enhance MRI images, whereas gas microbubbles can be employed to enhance ultrasound images. Fe_3_O_4_ NPs/polymer hybrid microbubble assemblies have been used for dual enhancement of MRI and USI (Figure 5a) [44]. Initially, cationic polyallylamine and trisodium citrate interact ionically to form ionomer aggregates (ICPA). Subsequently, citric acid-coated Fe_3_O_4_ NPs are introduced into the solution to create Fe_3_O_4_ NPs/polymer hybrid self-assemblies (IPHA). Anion-containing polyacrylic acid is then added to the solution, where it absorbs the cationic polyallylamine chains within ICPA and surrounding Fe_3_O_4_ NPs to form micro-capsules. Finally, the internal space of microcapsules is filled with carbon tetradecafluoride (C_6_F_14_) gas to produce hybrid microbubbles. In vitro and in vivo experimental results have demonstrated that microcapsules containing Fe_3_O_4_ NPs exhibit superior r_2_ relaxation properties compared to the monodispersed Fe_3_O_4_ NPs. This enhancement was attributed to the hybrid microcapsule aggregates, which provided Fe_3_O_4_ NPs with a higher relative volume fraction and magnetic susceptibility. Conversely, the hybrid microbubbles filled with gas inside displayed lower r_2_ relaxation performance than that of Fe_3_O_4_ NPs. This decline may result from the reduced interaction between Fe_3_O_4_ NPs in the shell and surrounding water. Once the microcapsule cavity is filled with gas, this leads to a decrease in local field inhomogeneity. Nevertheless, the gas-containing microbubbles demonstrated excellent T_2_-weighted MRI enhancement and USI capabilities. This approach allows for the control of microbubble size by adjusting the molar ratio of precursors, thereby modulating the imaging ability of the microbubbles.

The hybrid assembly, consisting of a core made of Fe_3_O_4_ NPs and a shell formed from biodegradable stearic acid-modified polyethylenimine (referred to as Stearic-LWPEI-Fe_3_O_4_ NPs), functions as an MRI-visible gene delivery system and can be utilized for MRI applications [83]. The primary effectiveness of this system is linked to its controllable cluster configuration, narrow size distribution, and ultra-sensitive imaging capabilities. This assembly is capable of forming nanocomplexes with microcircle DNA (mcDNA), which protects mcDNA from degradation and aids its internalization into cells via endocytosis. Cells that were transfected displayed reduced signal intensity in T_2_-weighted images, a phenomenon attributed to the shortened relaxation time of water molecules. Additionally, the contrast in signals between transfected and untransfected cells improved as the ratio of Stearic-LWPEI-SPIO/mcDNA nanocomplexes (N/P) increased. As the N/P ratio went up, the process of endocytosis became more pronounced, resulting in the enhanced signal contrast. In an in vivo mouse model, transfected cells incorporating the nanocomplexes were distinctly visible through MRI, indicating that these nanocomplexes can increase luciferase expression in MCF-7 cells without causing significant cytotoxic effects. Moreover, T_2_-weighted imaging indicated that the transfected cells showed more pronounced signal contrast compared to untreated cells. This research underscores the considerable potential of nanocomplexes for use as MRI-visible therapeutic diagnostics aimed at the delivery of small circular plasmid DNA.

### 3.2. Cancer Diagnosis and Therapy

The magnetic hyperthermia effect of Fe_3_O_4_ NP assemblies presents a promising strategy for cancer therapy [87,88]. The comparative analysis of hyperthermia effects between isolated Fe_3_O_4_ NPs and nanoclusters (NCs) of varying sizes revealed that the optimal NCs (25 nm) demonstrated a significantly enhanced heat generation efficiency compared to that of isolated Fe_3_O_4_ NPs (ca. 7 nm) [87]. Variations in the size of nanoclusters resulted in alternation to their interparticle crystalline structure, thereby influencing their magnetic and hyperthermia properties. The hyperthermia effect of NCs has been shown to effectively induce apoptosis of skin cancer cells.

Nanotechnology holds significant promise in the realm of tumor diagnosis and treatment, with photothermal methods being among the most frequently employed approaches. Fe_3_O_4_ NPs are regarded as safe materials for the clinical applications, however, their low molar extinction coefficient results in suboptimal photothermal performance. To enhance this performance, Fe_3_O_4_ NPs are often incorporated into polymers that exhibit high photothermal effects. Polydopamine (PDA) is one such polymer known for its excellent biocompatibility and photothermal conversion efficacy, making it suitable for improving the photothermal properties of Fe_3_O_4_ NPs (Figure 5b) [45]. Additionally, the self-assembled structure of Fe_3_O_4_ NPs can further augment their photothermal conversion capabilities. The Fe_3_O_4_@PDA superparticle consists of pre-assembled Fe_3_O_4_ superparticles as cores, subsequently coating them with PDA. These superparticles with magnetic properties could be directed to aggregate within tumors under the influence of external magnetic field, facilitating targeted diagnosis and treatment. By incorporating the targeting ligands such as folic acid, transferrin, and hyaluronic acid, the particles could bind to specific receptors on the surface of tumor cells, thereby enhancing both the targeting efficiency and biocompatibility of Fe_3_O_4_@PDA superparticles. Results indicated that the molar extinction coefficient of the assembled Fe_3_O_4_ superparticles was three orders of magnitude greater than that of individual Fe_3_O_4_ NPs. In vitro experiments demonstrated that Fe_3_O_4_@PDA superparticles exhibited remarkable photothermal conversion performance and biocompatibility. Furthermore, in vivo studies suggest that Fe_3_O_4_@PDA superparticles possess substantial potential for photothermal treatment of tumors.

Nanoparticle-based diagnostic and therapeutic integrative systems are emerging as a promising strategy for cancer treatment. These systems offer a dual functionality—diagnosing cancer while simultaneously delivering therapeutic agents. However, several challenges hinder their effective clinical application. Key among these challenges are the issues of diagnostic sensitivity, treatment efficiency, and the bioavailability of the nanoparticles. Additionally, the inherent heterogeneity of tumors, coupled with their capacity for drug resistance, complicates the treatment landscape, making it difficult to achieve consistent patient responses. In addressing these challenges, Hyeon and colleagues reported the innovative creation of pH-sensitive magnetic nanogrenades (PMNs) through the self-assembly of Fe_3_O_4_ NPs combined with pH-responsive ligands [86]. This unique design leverages a two-stage pH activation mechanism that facilitates a reversal of surface charge in the tumor microenvironment. This alteration not only enhances the adsorption of the particles to cancer cells but also improves their permeability. Furthermore, the endo/lysosomal pH-dependent therapeutic and diagnostic functionalities of PMNs are optimized through this design. When these PMNs come into contact with the acidic microenvironment characteristic of tumors, they undergo a transformation that shifts their surface charge to a positive state. This charge change significantly boosts their internalization efficiency within cancer cells. Upon reaching more acidic conditions present within lysosomes, PMNs disassemble into a highly active form, which activates their specific T_1_ magnetic resonance (MR) contrast and fluorescence capabilities. This pH-sensitive assembly and disassembly mechanism allows for enhanced MRI and increased photoactivity of the PMNs, specifically in the tumor’s acidic pH environment. Consequently, the innovative pH-responsive features of these nanoparticles effectively tackle two critical hurdles in cancer therapy: they enable early-stage diagnosis of tumors and facilitate treatment strategies for the heterogeneous and drug-resistant nature of cancerous tissues.

### 3.3. Drug Delivery

The mononuclear phagocyte system (MPS) is vital in identifying and eliminating foreign particles, which poses a considerable obstacle to nanoparticle delivery methods. In an effort to improve delivery efficiency, scientists have been investigating nanoplatforms featuring stealthy long loops. An innovative antiphagocytic delivery mechanism exhibiting “active” stealth characteristics was developed to enhance MRI performance alongside better drug delivery for cancer therapy [89]. They incorporated a self-peptide into biodegradable poly(lactic-glycolic acid)-poly(ethylene glycol) (PLGA-PEG), employing the self-assembly of PLGA-PEG and Fe_3_O_4_ NPs to create nanomicelles that encapsulate anticancer agent paclitaxel (PTX), thus forming an “active stealth” nanocarrier. The self-peptide interacts with the SIRP α receptor on macrophage surfaces, resulting in prolonged circulation time in the bloodstream and improved delivery effectiveness. Unlike the “passive” stealth properties offered by PEG or zwitterionic polymers, this “active self” nanomicelle successfully reduced MPS-mediated immune clearance and alleviated the accelerated blood clearance effect. Additionally, the synthesized nanomicelles exhibited remarkable T_2_ MRI contrast enhancement and effectively inhibited solid tumor growth. As a result, this technology presents a viable and widely applicable bottom-up approach for creating efficient antiphagocytic delivery systems.

Nanovesicles, which include liposomes and polymer vesicles, represent a category of assemblies characterized by an internal hollow structure capable of encapsulating and transporting both hydrophilic and hydrophobic drugs. These vesicles offer a unique advantage in drug delivery systems as they can release their carried drugs at specific target sites upon receiving external stimulation. This functionality has led to their widespread usage in drug encapsulation and release applications, where the precise delivery often translates into more effective treatment outcomes. The incorporation of superparamagnetic Fe_3_O_4_ NPs into the membranes of organic vesicles augments the magnetic responsiveness of these assemblies, thereby significantly enhancing their utility for tumor-targeted imaging and therapeutic delivery. Research conducted by Nie and collaborators highlighted the potential of innovative magnetic nanovesicles (MVs) featuring adjustable wall thickness, which can be effectively utilized for MRI-guided drug delivery (Figure 5c) [80]. These MVs exhibit superior magnetization levels compared to standalone Fe_3_O_4_ NPs, leading to the increased transverse relaxation rates and consequently darker MRI signals that facilitate imaging. Moreover, the anticancer drug doxorubicin (Dox) can be successfully encapsulated within the cavities of these multifaceted MVs, with the release of the Dox being modulated by altering the thickness of the vesicle membranes. The surface of multilayer-loaded MVs is further modified by conjugating it with the RGD peptide, which enhances their targeting capabilities. Upon intravenous injection, these multilayer nanovesicles exhibit a marked enrichment at the tumor site due to the combined effects of magnetic targeting and active targeting mediated by the RGD peptide. The findings of this study indicate a significant enhancement in MRI signals of magnetic nanovesicles compared to traditional strategies utilizing either magnetic or active targeting alone. This improvement correlates with increased drug delivery efficiency and heightened antitumor activity. This innovative nanoplatform showcases substantial promise for effective disease management by enabling the targeted delivery of imaging agents and therapeutics to hard-to-reach organs and tissues, surpassing the limitations posed by conventional delivery methods.

## 4. Summary and Outlook

In conclusion, self-assembly of Fe_3_O_4_ NPs represents a crucial area of exploration within the broader domains of nanoscience and nanotechnology. This phenomenon is influenced by a variety of forces, including magnetic dipole–dipole interactions, van der Waals forces, electrical double-layer steric interactions, and hydrophobic and solvation forces. By strategically employing surface ligands with distinct chemical structures, researchers have the ability to meticulously calibrate the interactions that govern the balance between attraction and repulsion among the particles. This control facilitates the emergence of various dimensional assemblies, encompassing 1D nanoparticle chains, 2D arrays, and 3D superstructures. The unique supermagnetic characteristics of Fe_3_O_4_ NPs enable researchers to manipulate self-assembly processes through the application of an external magnetic field, a feature that is not available with non-magnetic nanoparticles. This manipulation capability provides a significant advantage in controlling the arrangement and organization of nanoparticles during the self-assembly process. Moreover, specific surface ligands can play an important role in enhancing the water dispersibility of these nanoparticle assemblies. Current strategies predominantly emphasize the utilization of polymeric ligands with amphiphilic properties, which effectively stabilize the assemblies in aqueous environments and promote self-assembly via hydrophobic interactions. However, the choice of surface ligands is often constrained, limiting the potential for innovation in this area. Given that catechol termini exhibit a strong affinity for the surfaces of Fe_3_O_4_ NPs, the development of naturally occurring or synthetic catechol-terminated polymers and small ligands, which possess adjustable interaction capabilities, is a topic of considerable interest. Such advancements could facilitate the assembly of Fe_3_O_4_ NPs into intricate superstructures. Furthermore, there is a pressing need for the development of new techniques aimed at monitoring the self-assembly process in solution. These techniques would significantly contribute to a deeper understanding of the mechanisms underlying the formation of these nanoparticle assemblies.

Compared to dispersed Fe_3_O_4_ NPs, the assemblies showed significant enhancements in magnetic response. The good water dispersibility and improved magnetic properties of these assemblies make them suitable for more advanced biological applications, including imaging, cancer therapy and diagnosis, and drug delivery.

## Figures and Tables

**Figure 1 molecules-29-04160-f001:**
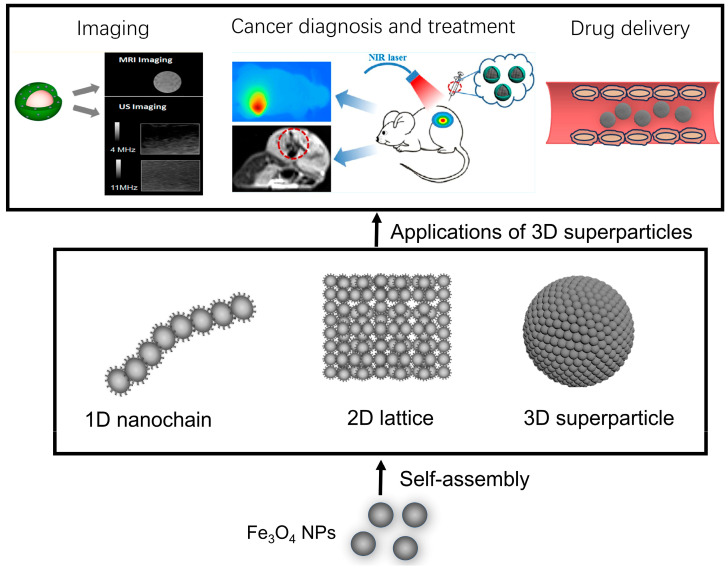
Diagrammatic representation of the primary topics covered in this review. The self-assembly of Fe_3_O_4_ NPs and their various applications. Taken with permission from [44,45].

**Figure 2 molecules-29-04160-f002:**
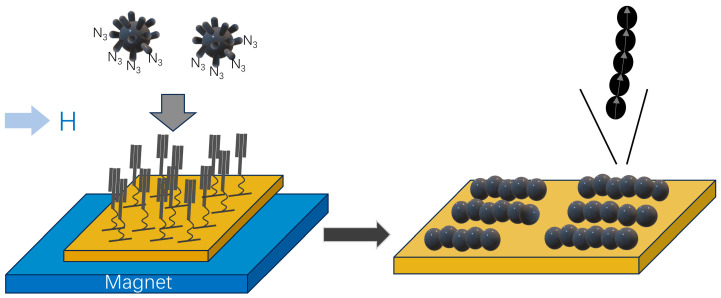
Schematic presentation for the preparation of 1D nanochains.

**Figure 3 molecules-29-04160-f003:**
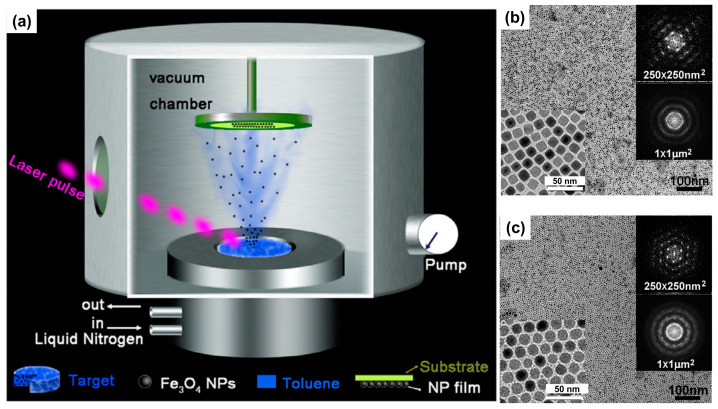
Schematic presentation (**a**) for the preparation of 2D lattice. TEM images (**a**,**b**) of the 2D lattice formed by nanocubes (**b**) and nanospheres (**c**). Taken with permission from [61].

**Figure 4 molecules-29-04160-f004:**
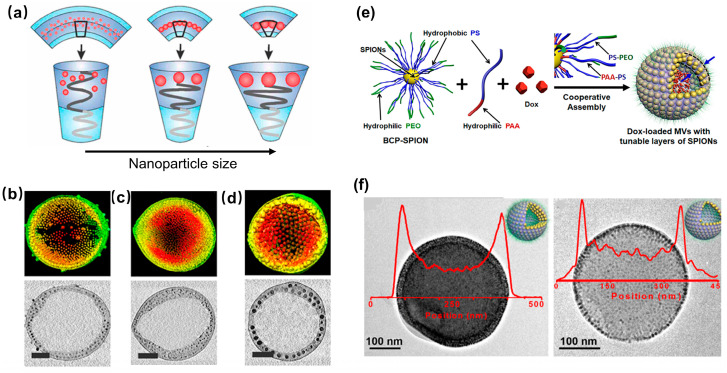
(**a**,**e**) Illustration for the preparation of Fe_3_O_4_ NVs. (**b**–**d**) Electron tomography data for Fe_3_O_4_ NVs. (**f**) STEM image and Fe intensity line scan for the multilayered and monolayered MVs. Taken with permission from [78,80].

**Figure 5 molecules-29-04160-f005:**
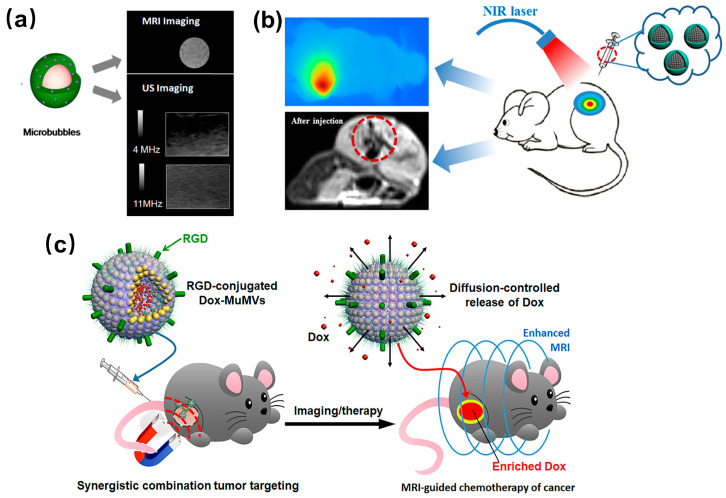
Schematic presentation of the applications of Fe_3_O_4_ NP assemblies in imaging (**a**), cancer diagnosis and treatment (**b**), and drug delivery (**c**). Taken with permission from [44,45,80].

## Data Availability

Data are contained within the article.

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
