# Peer review of "Magnetite Nanoparticle Assemblies and Their Biological Applications: A Review"

_molecules, 2024, doi:10.3390/molecules29174160_

Round 1
Reviewer 1 Report
Comments and Suggestions for Authors
The manuscript entitled “Recent advances of magnetite nanoparticle assemblies and their biological applications”, authored by Wei et al. provides a review on magnetite-based assemblies while addressing some key biological applications. The material included in the submitted version of the manuscript is worth of publication and potentially can attract the attention of the Journal’s readership interested in exploring the novelties of nanomagnetic-based assemblies in different dimensions, namely 1D, 2D, and 3D. However, the present version of the manuscript requires improvements in regard to its presentation, technical issues, and reference list as commented below. Therefore, the authors should prepare a revised version of the manuscript, either incorporating or rebutting (one by one) the comments herein included, while uploading the revised version plus a comprehensive cover letter file.
1. In terms of presentation, the main text fails while using technically inappropriate words (e.g. vital versus KEY; beside versus BESIDES; enhanced versus DISTINCTIVE; efficiency versus EFFICACY), surplus words (e.g. an; the), missing words (e.g. the), inappropriate usage words (e.g. while versus WHEREAS; impede versus HINDER). In this regard, it is required a careful revision of the entire text, captions, and legends.
2. In the Introduction section, very first line, the authors list “ferromagnetic materials” and forgot to include “ferrimagnetic materials”, the latter being the magnetic ordering of magnetite, the focus of the submitted review. A couple of sentences should be included in the first paragraph of the Introduction section explaining the differences between the two long range magnetic ordering. In this issue, a key reference should be added as well. This point should be taken into account while producing the revised version of the manuscript.
3. Still, in the first paragraph of the Introduction section, the authors exemplify the natural occurrence of magnetite nanoparticles (NPs) in living organisms, starting with the very simple (bacteria) and ending more complex ones (fish and birds). Actually, magnetite NPs has been found in a wider range, in the whole scale range of living beings; from unicellular (e.g. magnetic bacteria) up to complex mammals (e.g. humans). A very interesting example of the mid-range of living organisms are the migratory ants (e.g. Pachycondyla marginata), in which nanomagnetite-based 1D arrangement is used as compass for navigation while escaping from predators and searching for food. In this regard, the authors should check the literature for reports on this issue using paramagnetic resonance. For the benefit of the Journal’s readership, the authors should review the text in order to include the migratory ants example, adding extra references. This point should be considered while producing the revised version of the manuscript.
4. On sub-section 2.1 (1D Fe3O4 NPs nanoarrays), the authors did not include the very possibility of using externally-applied DC magnetic field to controllably produce 1D magnetite arrays from stable colloidal samples. Importantly, it should be emphasized in this approach, the influence of NP’s surface coating, NP’s volume fraction, NP’s average particles size, and the remarkable influence of the aging time in the process. Moreover, single chains of magnetite NPs can merge together building columnar-like nanostructures in a phase transition field-induced process, thus adding extra complexity and richness to the 1D nanoarrays. In this regard, the authors should check the literature for aging study of the field-induced columnar transition, which is remarkable demonstrated using magnetotransmissivity measurements. This point should be included in the revised version of the manuscript while corresponding references should be added in the revised reference list. This task should be performed before uploading the revised version of the manuscript.
5. On sub-section 2.2 (2D Fe3O4 NP lattice), the authors did not include a different approach for fabricating 2D magnetite-based lattice. This approach includes reconstruction of pre-fabricated 2D films comprising iron oxide (e.g. maghemite) NPs deposited onto flat surfaces. The paper published in the Scientific Reports, vol. 6, p. 18202, 2016 (doi: 10.1038/srep18202) explores the reconstruction of 2D arrays comprising maghemite NPs immersed into dimercaptosuccinic acid (DMSA) solutions at increasing molar concentration. Atomic force microscopy (AFM) micrographs provide strong evidence of different mechanisms for 2D reconstruction, depending on the DMSA concentration and aging time. To the benefit of the Journal’s readership, the authors should include the above-described approach of producing single layers of magnetic nanoparticles. The above-mentioned reference may be considered for inclusion in the revised reference list. This point should be considered while producing the revised version of the manuscript.
6. Still, on sub-section 2.2 (2D Fe3O4 NP lattice), which should be renamed as “2D Fe3O4 NP nanoarrays”, third paragraph, the authors mentioned “the self-assembly of Fe3O4 nanocubes featuring a hydrophobic surface resulted in the formation of flux-closure polygons”. In regard to self-assembly of single magnetic NPs producing 2D magnetite-based nanoarrays, it has been recently reported fabrication of magnetite-based nanorings. It is worth mentioning that, the newly-described magnetite nanorings displayed perfect magnetic vortex configuration, which may impact positively biomedical applications. For the benefit of the Journal’s readership, the authors should review the text in order to include the magnetite-based nanorings topic, adding extra references.
7. While addressing a review on magnetite-based nanostructures including the topic of sub-section 3.2 (Cancer diagnose and therapy), it is unexpectedly missing the magnetic hyperthermia effect while including only the photothermal effect as a promising cancer therapy. Actually, magnetite-based nanostructures are more suitable for the former than for the latter therapy modality. In this regard, the already mentioned report on magnetite-based nanorings includes the impressive magnetic hyperthermia performance evaluation. Moreover, an impressive magnetic hyperthermia performance and operating temperature capability control provide by core/shell (magnetic-NP/Au) has been recently reported in the journal Nanomaterials, vol. 12, p. 2760, 2022 (doi: 10.3390/nano12162760). To better inform the Journal's readership, the authors are requested to include a paragraph describing the magnetic hyperthermia using nanosized iron-oxide phases (including magnetite) and their wide application in cancer treatment. The authors are invited to check the above-mentioned publication and consider its inclusion in the revised version of the manuscript. This point should be considered before uploading the revised version of the manuscript.
Indeed, I found the manuscript interesting but there are many gaps to be bridged, as indicated above, in terms of presentation, data discussion, and reference list, that need to be addressed before this reviewer feels confident to support the manuscript’s publication.
Comments on the Quality of English Language
Minor revision required, as included in my review report (comments for Authors).
Author Response
Response to the Comments of Reviewer 1
Journal: Molecules
Manuscript ID: molecules-3160122
Type of manuscript: Review
Title: Recent Advances of Magnetite Nanoparticle Assemblies and Their Biological Applications
Author(s): Jinjian Wei*, Hong Xu, Yating Sun, Ying-Chun Liu, Ran Yan, Yuqin Chen*, Zhide Zhang
We would like to thank the reviewers for their valuable comments and suggestions. To address these comments and suggestions we have edited and revised the initial text. Please find below our point-by-point responses and revisions according to the reviewers' comments.
Reviewer 1
General comments: The manuscript entitled “Recent advances of magnetite nanoparticle assemblies and their biological applications”, authored by Wei et al. provides a review on magnetite-based assemblies while addressing some key biological applications. The material included in the submitted version of the manuscript is worth of publication and potentially can attract the attention of the Journal’s readership interested in exploring the novelties of nanomagnetic-based assemblies in different dimensions, namely 1D, 2D, and 3D. However, the present version of the manuscript requires improvements in regard to its presentation, technical issues, and reference list as commented below. Therefore, the authors should prepare a revised version of the manuscript, either incorporating or rebutting (one by one) the comments herein included, while uploading the revised version plus a comprehensive cover letter file.
Answer to general comments: Thank you very much for your comments. We revised the manuscript point-to-point according to your valuable comments.
Comment 1: In terms of presentation, the main text fails while using technically inappropriate words (e.g. vital versus KEY; beside versus BESIDES; enhanced versus DISTINCTIVE; efficiency versus EFFICACY), surplus words (e.g. an; the), missing words (e.g. the), inappropriate usage words (e.g. while versus WHEREAS; impede versus HINDER). In this regard, it is required a careful revision of the entire text, captions, and legends.
Answer to comment-1: Thank you very much for your comment. We carefully revised the main text according to your comment and marked the revisions as red in the manuscript.
Comment 2: In the Introduction section, very first line, the authors list “ferromagnetic materials” and forgot to include “ferrimagnetic materials”, the latter being the magnetic ordering of magnetite, the focus of the submitted review. A couple of sentences should be included in the first paragraph of the Introduction section explaining the differences between the two long range magnetic ordering. In this issue, a key reference should be added as well. This point should be taken into account while producing the revised version of the manuscript.
Answer to comment-2: Thank you very much for your comment. According to your comment, we include ferrimagnetic materials and carefully revised the main text as below.
Original: Magnetic nanoparticles (MNPs) are usually categorized as ferromagnetic materials, which can be elemental [1-4], alloy [5], oxide [6], or composite forms of substances primarily comprising iron (Fe), cobalt (Co), and nickel (Ni) [7]. In recent years, these materials have garnered significant attention due to their various applications, particularly in fields such as imaging [8], cancer therapy [9], and drug delivery [10,11], among others [12,13]. Among the different types of MNPs, magnetite nanoparticles (Fe3O4 NPs) stand out due to their excellent biocompatibility, which allows them to integrate seamlessly into biological systems.
Revision: Magnetic nanoparticles (MNPs) are usually categorized as ferromagnetic and ferrimagnetic materials. The composition of ferromagnetic nanoparticles can be elemental [1-4], alloy [5], oxide [6], or composite forms of substances primarily comprising iron (Fe), cobalt (Co), and nickel (Ni) [7]. In contrast to ferromagnetic substances, ferrimagnetic materials are typically dual oxides composed of iron and another metal [8]. The ferrimagnetic materials most commonly used in technical devices are known as ferrites. These ferrites are cubic or hexagonal shape. Cubic ferrites have a chemical formula of MO·Fe2O3, where M represents a divalent ion such as Mn2+, Fe2+, Co2+, or Ni2+. In recent years, these materials have garnered significant attention due to their various applications, particularly in fields such as imaging [9], cancer therapy [10-12], and drug delivery [13,14], among others [15,16]. Among the different types of MNPs, magnetite nanoparticles (Fe3O4 NPs) stand out due to their excellent biocompatibility, which allows them to integrate seamlessly into biological systems.
Comment 3: Still, in the first paragraph of the Introduction section, the authors exemplify the natural occurrence of magnetite nanoparticles (NPs) in living organisms, starting with the very simple (bacteria) and ending more complex ones (fish and birds). Actually, magnetite NPs has been found in a wider range, in the whole scale range of living beings; from unicellular (e.g. magnetic bacteria) up to complex mammals (e.g. humans). A very interesting example of the mid-range of living organisms are the migratory ants (e.g. Pachycondyla marginata), in which nanomagnetite-based 1D arrangement is used as compass for navigation while escaping from predators and searching for food. In this regard, the authors should check the literature for reports on this issue using paramagnetic resonance. For the benefit of the Journal’s readership, the authors should review the text in order to include the migratory ants example, adding extra references. This point should be considered while producing the revised version of the manuscript.
Answer to comment-3: Thank you very much for your important comments. According to your comments, we included the migratory ant and extra references. We carefully revised the manuscript as below.
Original: They are naturally found in living organisms such as magnetic bacteria [14,15], fish [16], and birds [17], which enhances their appeal for biomedical applications.
Revision: They are naturally found in living organisms such as magnetic bacteria [17,18], migratory ants [19,20], fish [21], and birds [22]. A very interesting example of the mid-range of living organisms are migratory ants, in which nanomagnetite-based one dimensional arrangement is used as compass for navigation while escaping from predators and searching for food. These enhance their appeal for biomedical applications.
Comment 4: On sub-section 2.1 (1D Fe3O4 NPs nanoarrays), the authors did not include the very possibility of using externally-applied DC magnetic field to controllably produce 1D magnetite arrays from stable colloidal samples. Importantly, it should be emphasized in this approach, the influence of NP’s surface coating, NP’s volume fraction, NP’s average particles size, and the remarkable influence of the aging time in the process. Moreover, single chains of magnetite NPs can merge together building columnar-like nanostructures in a phase transition field-induced process, thus adding extra complexity and richness to the 1D nanoarrays. In this regard, the authors should check the literature for aging study of the field-induced columnar transition, which is remarkable demonstrated using magneto transmissivity measurements. This point should be included in the revised version of the manuscript while corresponding references should be added in the revised reference list. This task should be performed before uploading the revised version of the manuscript.
Answer to comment-4: Thank you very much for your comments. According to your comments, we added the discussion for aging study of field induced columnar transition. And we revised manuscript as below.
Original: This study demonstrates that magnetic field-induced self-assembly methods can be successfully employed to create polymer films exhibiting both macroscopic order and nanometer-scale structure, even under low magnetic field conditions. 1D nanoarrays can also be generated independent of external magnetic field [44].
Revision: This study demonstrates that magnetic field-induced self-assembly methods can be successfully employed to create polymer films exhibiting both macroscopic order and nanometer-scale structure, even under low magnetic field conditions. In another report, 1D Fe3O4 NP arrays was controllably generated using tartrate- and polyaspartate-coated colloidal Fe3O4 NPs through the application of external magnetic field [51]. It was found that larger Fe3O4 NPs enhance the magnetic dipole interaction when compared with their smaller counterparts. Polyaspartate, which is longer than tartrate surfactant molecules, assure the stability with smaller aggregates. Theoretical analysis suggested that the average chain length tends to rise as both the aging time and particle volume fraction increase. As the strength of applied magnetic field increases, single chains of Fe3O4 NPs can merge together building columnar-like nanostructures in a phase transition field-induced process, thus adding extra complexity and richness to the 1D nanoarrays. 1D nanoarrays can also be generated independent of external magnetic field [49].
Comment 5: On sub-section 2.2 (2D Fe3O4 NP lattice), the authors did not include a different approach for fabricating 2D magnetite-based lattice. This approach includes reconstruction of pre-fabricated 2D films comprising iron oxide (e.g. maghemite) NPs deposited onto flat surfaces. The paper published in the Scientific Reports, vol. 6, p. 18202, 2016 (doi: 10.1038/srep18202) explores the reconstruction of 2D arrays comprising maghemite NPs immersed into dimercaptosuccinic acid (DMSA) solutions at increasing molar concentration. Atomic force microscopy (AFM) micrographs provide strong evidence of different mechanisms for 2D reconstruction, depending on the DMSA concentration and aging time. To the benefit of the Journal’s readership, the authors should include the above-described approach of producing single layers of magnetic nanoparticles. The above-mentioned reference may be considered for inclusion in the revised reference list. This point should be considered while producing the revised version of the manuscript.
Answer to comment-5: Thank you very much for your comments. We carefully read your recommended reference. However, there is no clear discussion about the construction of single layer of magnetic nanoparticles. According to your comment, we added the recommended reference, and revised the manuscript as below.
Original: This selective nucleation capability enables the formation of intricate nanoparticle chains and arrays that can be tailored to the geometry of underlying patterns, allowing for the creation of chain formations that may include shapes such as rings or squares. In a recent study, the self-assembly of Fe3O4 nanocubes featuring a hydrophobic surface resulted in the formation of flux-closure polygons via a drop-casting method [60].
Revision: This selective nucleation capability enables the formation of intricate nanoparticle chains and arrays that can be tailored to the geometry of underlying patterns, allowing for the creation of chain formations that may include shapes such as rings or squares. Morais and coworkers reported the reconstruction of pre-fabricated 2D films comprising iron oxide (e.g. maghemite) NPs deposited onto flat glass surfaces [65]. The reconstruction of 2D arrays of maghemite NPs immersed into dimercaptosuccinic acid (DMSA) solutions at increasing molar concentration was investigated. Atomic force microscopy micrographs provide strong evidence of different mechanisms for 2D reconstruction, depending on the DMSA concentration and aging time. In a recent study, the self-assembly of Fe3O4 nanocubes featuring a hydrophobic surface resulted in the formation of flux-closure polygons via a drop-casting method [67].
Comment 6: Still, on sub-section 2.2 (2D Fe3O4 NP lattice), which should be renamed as “2D Fe3O4 NP nanoarrays”, third paragraph, the authors mentioned “the self-assembly of Fe3O4 nanocubes featuring a hydrophobic surface resulted in the formation of flux-closure polygons”. In regard to self-assembly of single magnetic NPs producing 2D magnetite-based nanoarrays, it has been recently reported fabrication of magnetite-based nanorings. It is worth mentioning that, the newly-described magnetite nanorings displayed perfect magnetic vortex configuration, which may impact positively biomedical applications. For the benefit of the Journal’s readership, the authors should review the text in order to include the magnetite-based nanorings topic, adding extra references.
Answer to comment-6: Thank you very much for your important comments. According to your comment, we renamed the title of subsection 2.2 as 2D Fe3O4 NP nanoarrays. In section 2.2, we focused on the fabrication of 2D nanoarrays based on the self-assembly of NPs, not the morphology of NPs. As you mentioned, the reference [Materials & Design 232 (2023) 112082] reported the fabrication of magnetite-based nanorings and their magnetothermal evaluations. However, there is no discussion on self-assembly of nanorings into 2D nanoarrays. Based on these, we think we should not make discussion on this reference in subsection 2.2. However, we cite this paper in the Introduction section.
Comment 7: While addressing a review on magnetite-based nanostructures including the topic of sub-section 3.2 (Cancer diagnose and therapy), it is unexpectedly missing the magnetic hyperthermia effect while including only the photothermal effect as a promising cancer therapy. Actually, magnetite-based nanostructures are more suitable for the former than for the latter therapy modality. In this regard, the already mentioned report on magnetite-based nanorings includes the impressive magnetic hyperthermia performance evaluation. Moreover, an impressive magnetic hyperthermia performance and operating temperature capability control provide by core/shell (magnetic-NP/Au) has been recently reported in the journal Nanomaterials, vol. 12, p. 2760, 2022 (doi: 10.3390/nano12162760). To better inform the Journal's readership, the authors are requested to include a paragraph describing the magnetic hyperthermia using nanosized iron-oxide phases (including magnetite) and their wide application in cancer treatment. The authors are invited to check the above-mentioned publication and consider its inclusion in the revised version of the manuscript. This point should be considered before uploading the revised version of the manuscript.
Answer to comment-7: Thank you very much for your important comments. We agree with you that hyperthermia effects is the most important application of magnetic nanoparticles. We are sorry that we cannot include the two references [Materials & Design 232 (2023) 112082; Nanomaterials, 2022, 12, 2760] in sub-section 3.2, since this section introduces the application of assemblies of Fe3O4 NPs. However, we added the above two references into the applications of magnetic nanoparticles in the Introduction section. On the other hand, we find two references [ACS Omega 2021, 6, 31161−31167; Sci. Rep. 2013, 3, 1652] related to the hyperthermia effects of assemblies of Fe3O4 NPs. Thus, we include the above references into sub-section 3.2 to complete the introduction of hyperthermia effects and discussed as below.
Revision: The magnetic hyperthermia effect of Fe3O4 NP assemblies presents a promising strategy for cancer therapy [89,90]. The comparative analysis of hyperthermia effects between isolated Fe3O4 NPs and nanoclusters (NCs) of varying sizes revealed that the optimal NCs (25 nm) demonstrated a significantly enhanced heat generation efficiency compared to that of isolated Fe3O4 NPs (ca. 7 nm) [89]. Variations in the size of nanoclusters resulted in alternation to their interparticle crystalline structure, thereby influencing their magnetic and hyperthermia properties. The hyperthermia effect of NCs has been shown to effectively induce apoptosis of skin cancer cells.
Reviewer 2 Report
Comments and Suggestions for Authors
In the article, the assemblies of magnetite nanoparticles and their biological applications were described. The authors focused more on medical applications such as imaging, cancer treatment and diagnosis, and drug delivery. After some major changes, the manuscript could be a great addition to the journal. I recommend the publication of this work after the addressing of the following comments:
Major:
- The scope of the review is missing. In the title information about recent advances suggests that the newest publication will be cited. However, in the references, we see only 7 publications that are not older than 5 years. What criteria were used to choose such a small set of recent publications?
- Explain why 12nm is the limit for a static dephasing regime. Is this value independent of other factors like frequency, and distance between nanoparticles?
- In the article: "In comparison to isolated Fe3O4 nanoparticles (NPs), assemblies of Fe3O4 NPs exhibit a significantly enhanced magnetic moment, which in turn results in improved sensitivity for detection purposes [12,26-31]. " Explain what it means. How magnetic moment can be enhanced? What do you mean by improved sensitivity? Is the signal better for assemblies than separated nanoparticles with the same mass concentration of magnetite? What information is taken from which references?
Minor:
- In the abstract, the authors highlight the unique superparamagnetic properties of magnetite nanoparticles. Other magnetic nanoparticles also have superparamagnetic properties. The unique adjective is misleading.
- In the Introduction the authors talk about larger nanoparticles in the range of 4 to 12 nanometers tend to have higher lateral relaxation rate. This sentence is hard to understand.
- The last section "Summary and Outlook" has the number 5 when it should be 4.
- Figure 1 is confusing. Does this mean that all 1D, 2D, and 3D structures have all mentioned applications? The image shows applications only for 3D superparticles.
- In the article: "Compared with individual Fe3O4 NPs, their assemblies showed the enhanced magnetic response, functionality and water-dispersibility". This sentence is to vege. What does it mean "magnetic response", "functionality", or "water-dispersibility"? How did just making assemblies enhance these properties? Maybe some other factors like surface functionalization influence water dispersibility and this can also be achieved for single nanoparticles.
Author Response
Response to the Comments of Reviewer 2
Journal: Molecules
Manuscript ID: molecules-3160122
Type of manuscript: Review
Title: Recent Advances of Magnetite Nanoparticle Assemblies and Their Biological Applications
Author(s): Jinjian Wei*, Hong Xu, Yating Sun, Ying-Chun Liu, Ran Yan, Yuqin Chen*, Zhide Zhang
We would like to thank the reviewers for their valuable comments and suggestions. To address these comments and suggestions we have edited and revised the initial text. Please find below our point-by-point responses and revisions according to the reviewers' comments.
Reviewer 2
General comments: In the article, the assemblies of magnetite nanoparticles and their biological applications were described. The authors focused more on medical applications such as imaging, cancer treatment and diagnosis, and drug delivery. After some major changes, the manuscript could be a great addition to the journal. I recommend the publication of this work after the addressing of the following comments.
Answer to general comments: Thank you very much for your comment. We carefully revised the main text according to your comment and marked the revisions as red in the manuscript.
Comment 1: The scope of the review is missing. In the title information about recent advances suggests that the newest publication will be cited. However, in the references, we see only 7 publications that are not older than 5 years. What criteria were used to choose such a small set of recent publications?
Answer to comment-1: Thank you very much for your important comment. We think recent 10 years are the criteria for recent advances. And we agree with your comment. Thus, we change the title to “Magnetite Nanoparticle Assemblies and Their Biological Applications: A Review”.
Comment 2: Explain why 12nm is the limit for a static dephasing regime. Is this value independent of other factors like frequency, and distance between nanoparticles?
Answer to comment-2: Thank you very much for your important comments. According to the reported references, for nanoparticles larger than 12 nm, the r2 relaxivity of nanoparticles does not continue to increase as the size does. This region is called static dephasing regime (SDR). In the SDR, the magnetic field generated by nanoparticles is so strong that diffusion of water has little influence on the T2 relaxation process. Therefore, a plateau of the maximum r2 is predicted to appear in SDR. To clarify this, we revised the manuscript as below.
Original: Interestingly, for nanoparticles exceeding the size of 12 nm, the r2 relaxivity does not continue to rise with increasing the size. This particular size region is referred to as the static dephasing regime (SDR) [24,25]. Beyond this specific threshold, the relaxivity begins to decrease as the size of the nanoparticles increases further.
Revision: Interestingly, for nanoparticles exceeding the size of 12 nm, the r2 relaxivity does not continue to rise with increasing the size. This particular size region is referred to as the static dephasing regime (SDR) [29,30]. In the SDR, the magnetic field produced by nanoparticles is sufficiently strong that the diffusion of water has minimal impact on the T2 relaxation process. Consequently, it is anticipated that a plateau in the maximum r2 will emerge in SDR. Beyond this specific threshold, the relaxivity begins to decrease as the size of nanoparticles increases further.
Comment 3: In the article: "In comparison to isolated Fe3O4 nanoparticles (NPs), assemblies of Fe3O4 NPs exhibit a significantly enhanced magnetic moment, which in turn results in improved sensitivity for detection purposes [12,26-31]. " Explain what it means. How magnetic moment can be enhanced? What do you mean by improved sensitivity? Is the signal better for assemblies than separated nanoparticles with the same mass concentration of magnetite? What information is taken from which references?
Answer to comment-3: Thank you very much for your important comments. To better understand the enhancement of magnetic moment of Fe3O4 NP assemblies, we take clusters as an example. A magnetic nanoparticle cluster can theoretically be regarded as a large magnetized sphere, and its overall magnetic moment is proportional to the size [J. Magn. Magn. Mater 293 (2005) 532–539; Chem. Rev. 2015, 115, 10637−10689]. Therefore, the clusters of Fe3O4 NPs exhibited a higher magnetic moment compared with isolated Fe3O4 NPs. The enhancement of magnetic moment in the 1D or 2D nanostructures can similarly be explained, while the transverse relaxivity of nanoparticle assemblies depends on their magnetization [Chem. Rev. 2015, 115, 10637−10689]. The literature also reported that assemblies are distinctive in relaxivity because their peculiar nanostructures change the proton relaxation effect [Adv. Mater. 2013, 25, 5196–5214]. It is demonstrated that the assemblies facilitate T2 relaxation process, displaying higher r2 relaxivity, and can be an effective strategy to increase the magnetic responsiveness [Adv. Mater. 2013, 25, 5196–5214]. Based on the above findings, the assemblies exhibit improved detection sensitivity of magnetic resonance imaging. To clarify this, we revised the manuscript as below.
Original: In comparison to the isolated Fe3O4 nanoparticles (NPs), assemblies of Fe3O4 NPs exhibit a significantly enhanced magnetic moment, which in turn results in improved sensitivity for detection purposes [12,26-31].
Revision: In comparison to the isolated Fe3O4 NPs, their assemblies possess unique relaxivity characteristics, as their specific nanostructures change the proton relaxation phenomenon[34]. Taking clusters as an example, a collection of magnetic nanoparticles can theoretically be visualized as a large sphere with a magnetic charge, where its total magnetic moment correlates with the size [15]. Consequently, Fe3O4 NP clusters demonstrate a greater magnetic moment in comparison to dispersed Fe3O4 NPs. While the transverse relaxivity of the assemblies is influenced by their magnetization levels [15]. In light of these observations, the assemblies offer enhanced sensitivity of detection in magnetic resonance imaging..
Comment 4: In the abstract, the authors highlight the unique superparamagnetic properties of magnetite nanoparticles. Other magnetic nanoparticles also have superparamagnetic properties. The unique adjective is misleading.
Answer to comment-4: Thank you very much for your important comment. According to your comment, we delete the word “unique”.
Comment 5: In the Introduction the authors talk about larger nanoparticles in the range of 4 to 12 nanometers tend to have higher lateral relaxation rate. This sentence is hard to understand.
Answer to comment-5: Thank you very much for your important comment. According to your comment, we changed the expression of your-mentioned sentence and revised as below.
Original: Experimental observations indicate that larger Fe3O4 NPs, particularly those within a size range of 4 to 12 nanometers, tend to have a higher lateral relaxation rate (r2) relaxivity compared to their smaller counterparts [23].
Revision: Experimental observations indicate that larger Fe3O4 NPs tend to have a higher lateral relaxation rate (r2) relaxivity in the size range of 4 to 12 nm [28].
Comment 6: The last section "Summary and Outlook" has the number 5 when it should be 4.
Answer to comment-6: Thank you very much for your important comment. We are sorry for this mistake. According to your comment, we change the number from 5 to 4.
Comment 7: Figure 1 is confusing. Does this mean that all 1D, 2D, and 3D structures have all mentioned applications? The image shows applications only for 3D superparticles.
Answer to comment-7: Thank you very much for your important comment. We are sorry for this misleading. We only present the applications of 3D superpartcles. To clarify this, we emphasized 3D superparticles in Figure 1.
Comment 8: In the article: "Compared with individual Fe3O4 NPs, their assemblies showed the enhanced magnetic response, functionality and water-dispersibility". This sentence is to vege. What does it mean "magnetic response", "functionality", or "water-dispersibility"? How did just making assemblies enhance these properties? Maybe some other factors like surface functionalization influence water dispersibility and this can also be achieved for single nanoparticles.
Answer to comment-8: Thank you very much for your important comment. According to your comment, we revised the manuscript as below.
Original: Compared with individual Fe3O4 NPs, their assemblies showed the enhanced magnetic response, functionality and water-dispersibility.
Revision: Compared with individual Fe3O4 NPs, their assemblies showed the enhanced magnetic response.
Round 2
Reviewer 1 Report
Comments and Suggestions for Authors
The revised version of the manuscript satisfactorily includes all the points and concerns I have included in my previous report. Therefore, the manuscript should be accepted for publication.
Reviewer 2 Report
Comments and Suggestions for Authors
The authors have addressed all my comments. I recommend this article for publication.